# Does Mobile Internet Use Affect the Loneliness of Older Chinese Adults? An Instrumental Variable Quantile Analysis

**DOI:** 10.3390/ijerph19095575

**Published:** 2022-05-04

**Authors:** Zenghua Guo, Boyu Zhu

**Affiliations:** 1School of Marxism, Hubei University of Economics, Wuhan 430205, China; guozenghua@hbue.edu.cn; 2School of Sociology, Wuhan University, Wuhan 430072, China

**Keywords:** mobile internet, loneliness in the elderly, instrumental variable method, influence mechanism

## Abstract

Based on the 2018 China Family Panel Studies (CFPS) data, we empirically analyze the effect, heterogeneity, quantile differences and influencing mechanisms of mobile Internet use on loneliness in the elderly. The study found that the loneliness of the elderly who used mobile Internet was 33.1% lower than that of the elderly who did not use the mobile Internet The study also passed the robustness test. There is heterogeneity in the effect of mobile Internet use on loneliness among the elderly of different ages, educational levels and marital status. Among them, the use of mobile Internet has a significant alleviating effect on the loneliness of the 60–70-year-old elderly, those of junior high school education level and below, and the elderly with a partner. The quantile regression analysis showed that the elderly group with a high level of loneliness benefited more from the use of mobile Internet. Mediation analysis further showed that mobile Internet use can improve parent–child relationship, increase offline interactions with children, and reduce children’s tangible support, which we interpret as a potential mechanism for mobile Internet use to alleviate loneliness in the elderly.

## 1. Introduction

The world has entered a new era of aging populations, and China is also facing the challenge. Predicted in “China Development Report 2020: Development Trends and Policies of China’s Aging Population”, by 2022, China’s population over the age of 65 will account for 14% of the total population. In response to the challenges of the aging population, the World Health Organization in 1990 adopted “healthy aging” as a development strategy. Advocacy for healthy aging not only includes ensuring the physical health of the elderly, but also requires the elderly to maintain mental health and good social adaptability. Loneliness is an important indicator of the mental health of the elderly, and it may also adversely affect the physical health of the elderly. Previous studies have pointed out that loneliness develops in a “U-shaped” trend in an individual’s life course, rising in youth, decreasing in mid-adulthood, and increasing in old age [1,2]. According to the “Report on the Development of Quality of Life for the Elderly in China (2019)” by the Research Center for Aging Research, nearly one-third of the young elderly in China feel lonely, and more than half of the older elderly are lonely [3]. This shows that the middle-aged and elderly groups in Chinese society are currently facing a serious problem with loneliness, which is not in line with the advocacy of healthy aging and is not conducive to a society’s active response to the aging problem. Therefore, loneliness in the elderly and its influencing factors have attracted extensive attention from scholars.

At the same time, the current society has also entered a new era of digitalization. The Internet has gradually penetrated into all aspects of individual life and has an important impact on people’s lives. The elderly are living in a digital society, and the Internet has an increasing impact on them. The 48th “Statistical Report on China’s Internet Development Status” pointed out that the growth rate of middle-aged and elderly netizens is the fastest, with 28% of netizens over the age of 50, an increase of 5.2 percentage points in June 2020. The rapid development of the Internet has two sides for the elderly. On the one hand, the advancement of Internet technology can facilitate the lives of the elderly, and online medical treatment and online interaction can help the elderly overcome physical obstacles and obtain social support. On the other hand, due to a lack of digital literacy, the elderly often find it difficult to master digital technology, and there is a large digital gap between them and other groups. This gap reduces the self-efficacy of the elderly, and psychologically produces feelings of being abandoned by the times. Such negative cognition in the elderly is not conducive to their mental health. In the context of population aging and digital life, whether and how Internet use affects the sense of loneliness in the elderly has become a topic worthy of in-depth study.

This paper attempts to explore the impact of mobile Internet use on loneliness in the elderly and to further analyze the changes in the quantile distribution of loneliness. Given that previous studies have focused on the impact of other forms of ICT (such as general Internet use, WeChat, and other software) on loneliness in the elderly, research on the impact of mobile Internet use is very limited. Compared with computers, smartphones are more popular among the public due to their advantages of being cheap and portable. Even the elderly who are in a disadvantaged position in the information age can own devices such as smartphones. Therefore, mobile Internet use may have a greater impact on the lives of older age groups. This study is innovative in the following aspects and complements the existing literature. First, we used an instrumental variable approach to overcome potential endogeneity to test whether the effect of mobile Internet use on loneliness in the elderly is causal. Secondly, this paper uses an instrumental variable quantile regression model to explore whether the effect of mobile Internet use on loneliness in the elderly is different in different quantiles. Finally, we used the method of mediation analysis to explore and analyze the underlying mechanisms of the effect of mobile Internet use on loneliness in the elderly.

## 2. Prior Literature

The concept of loneliness originated from medical science, which represents the dysfunction of interpersonal communication and emotional expression, and then was introduced into social psychology [4]. Generally, loneliness is defined as people’s subjective unpleasant and painful emotional experience. This experience is generated when there is a quantitative and qualitative gap between the expected social relationship and the actually perceived social relationship [5]. That is, the subjective psychological feeling that the individual desires social interpersonal communication but feels alienated and rejected by the social system. Weiss further divides loneliness into social loneliness and emotional loneliness. Social loneliness refers to loneliness caused by the inability to meet the needs of social integration or lack of social sense. Emotional loneliness refers to the feeling of loneliness caused by the inability of individuals to meet their attachment needs [6]. Generally speaking, individuals feel more or less lonely in society, but long-term or severe loneliness affects the physical and mental health and normal lives of individuals. As far as the elderly are concerned, loneliness, as an important indicator to measure their mental health, is not only closely related to mental health conditions such as anxiety and depression, but is also a risk factor for increased mortality, hypertension, diabetes, cardiovascular disease, mobility, cognitive dysfunction, sleep disorders, and other physiological diseases in middle-aged and elderly people [7,8,9]. In the context of the aging population, because loneliness is closely related to health, it is of great significance to understand and study the problem of loneliness in the elderly. In view of the many adverse effects of loneliness on the elderly, its influencing factors have become the focus of a large number of studies.

Existing studies related to the topic of this study have focused on the relationship between Internet use and loneliness but have not reached a consistent conclusion on this issue. First, there are different views on the correlation and causality between the Internet and loneliness. Some scholars believe that loneliness affects Internet use behavior, and individuals with high levels of loneliness are more likely to use the Internet to obtain emotional support and regulate negative emotions [10]. Some scholars also believe that there is a causal relationship between Internet use and loneliness; that is, lonely people have online social preferences, and these preferences further lead to negative results related to Internet use [11]. Secondly, there are differences in the conclusions drawn by the existing studies on the issue of the impact of Internet use on loneliness, among which there are mainly two viewpoints: the substitution theory and the expansion theory. The substitution theory believes that Internet use occupies the interaction in the actual field space of the elderly, reduces individual offline social connection and social participation, and leads to an increase in individual loneliness [12,13]. However, some scholars pointed out that there are sample selection defects in the research of Kraut, and their sample was at risk of increased loneliness even if they did not use the Internet at this stage, and thus their conclusions may require further validation [14]. The expansion theory suggests that Internet use can alleviate individual loneliness. Scholars pointed out that the Internet, as a communication tool and entertainment platform, can help the elderly overcome the barriers of time and distance, bring about an increase in the frequency of social interaction, allow the elderly to expand or maintain social connections, and avoid social isolation while expanding access to information and enriching the lives of the elderly, thereby reducing the level of loneliness among the elderly [15,16,17]. In addition, different functions and durations of Internet use have different effects on loneliness. Research has shown that greater use of the Internet as a communication tool is associated with lower social loneliness, while greater use of the Internet to find new friends is associated with higher levels of emotional loneliness, and more time spent on the internet is associated with higher levels of social loneliness [18]. Finally, there are studies that believe that Internet use can promote the mental health of the elderly, but it does not play a significant role in relieving the loneliness of the elderly, and there is no difference in the level of loneliness between the elderly who use the Internet and others who use the Internet [19].

Although smartphones and mobile Internet have become an important part of people’s daily lives, few studies have empirically analyzed the impact of mobile Internet use on loneliness in the elderly in the Chinese context. Specifically, most of the existing studies focus on the effect of Internet and social media use on loneliness in general, and point out that Internet and social media use has a significant negative correlation with loneliness in the elderly. However, there are still different results in its impact on the loneliness of the elderly in different age groups. Some scholars believe that Internet use is more obvious in the younger age group of 60–69 years through empirical analysis [20], and it has no effect on the elderly over the age of 70, and as age increases, the effect of Internet use in reducing the loneliness of the elderly gradually weakens [21]. However, some scholars have pointed out that because middle-aged and elderly people face more unfavorable situations, the impact of Internet use on the middle elderly and older elderly is more obvious [22]. The research on the impact of mobile Internet use on loneliness focuses on adolescents. Empirical research shows that college students’ excessive use of mobile Internet and excessive reliance on mobile phones does not alleviate loneliness but may instead cause withdrawal from interpersonal relationships, thereby further enhancing loneliness [23]. However, there are many differences between the elderly group and the adolescent group in terms of physiological function, lifestyle, social interaction, etc. The reasons for the loneliness of the two are not exactly the same, so the effect of mobile Internet use on loneliness among adolescents may not be applicable to the elderly group. Therefore, the effect of mobile Internet use on loneliness in the elderly deserves further research.

Overall, most existing studies have focused on the correlation between Internet use and loneliness in older age. However, the existing research still has some limitations. First, the previous studies mostly focused on this research issue in the Western context, and the number of related studies carried out in the Chinese context was relatively small. Second, although more and more studies have focused on the relationship between Internet use and loneliness in the elderly, few empirical studies have explored the effect of mobile Internet use on the distribution of loneliness in the elderly through quantile regression. At the same time, the heterogeneity analysis of the influence of Internet use on loneliness in the elderly is insufficient, and the conclusions are different. Third, in order to overcome the endogeneity problem between the two and explore the causal relationship between the two, most studies use the propensity value matching method, and few studies use the instrumental variable method to study and test this issue. Finally, although studies have demonstrated that Internet use has a certain effect on loneliness in the elderly, few empirical studies have focused on the underlying mechanism behind the effect, that is, how Internet use affects the loneliness of the elderly.

## 3. Data and Methods

### 3.1. Data and Variable Selection

This paper employs data from the China Family Panel Studies (CFPS), a nationally representative, longitudinal survey launched in 2010 by the Institute of Social Science Survey of Peking University in China. The CFPS contains approximately 14,960 households from 25 provinces/municipalities in China (excluding Taiwan, Macao, Hong Kong, Tibet, Inner Mongolia, Xinjiang, Qinghai, Ningxia, and Hainan), representing about 95 percent of the Chinese population. Four follow-up surveys were conducted in 2012, 2014, 2016, and 2018. Because data on MIU is only available in the latest survey, we only use the 2018 survey for the analysis. Our study focuses on the relationship between MIU and loneliness in older adults over 60 years of age, and we exclude from the sample those who lacked information on core variables (including loneliness and MIU). As a result, 7128 observations were used for analysis after sample selection and data cleaning.

Our dependent variable is loneliness. Based on the available information in our data, the analysis is constructed based on respondents’ 4-point ordinal scale answer to the question “In the past week, how many days did you feel lonely?”, the options for which include 1 = less than a day, 2 = one or two days, 3 = three or four days, and 4 = five days or more, with a higher score indicating a higher level of loneliness. The histogram in Figure 1 demonstrates the distribution of loneliness and shows that the distribution is skewed to the left. It is worth noting that, compared with the study of Song et al. [24], the distribution of loneliness in this study is more skewed to the left. In particular, the mean loneliness score (1.486, measured on a 4-point scale) in our study was lower than that found by Xu et al. [25] (1.77, measured on a 4-point scale). However, it still needs to be noted that there is a considerable proportion of the elderly facing a negative psychological state with a high sense of loneliness. At the same time, the expression of emotions in Eastern culture tends to be introverted, and negative emotions are underestimated in daily presentations and questionnaires. Therefore, the true perception of loneliness in the elderly population may be shifted to the right in comparison with the sample presentation. In conclusion, the influencing factors and mechanisms of loneliness in the elderly still need to be further explored so as to improve the well-being and health status of the elderly and to achieve active aging.

Following Khalaila and Vitman-Schorr and Heo et al. [26,27], our independent variable, MIU, was operationalized as a dichotomous measure that equals to one if the respondent uses the mobile internet through any mobile devices (e.g., smartphone and iPad) in his/her daily life.

In addition, following the existing literature, we controlled for a range of demographic characteristics and covariates that maybe correlated with MIU and older adults’ loneliness. In this paper, variables such as gender, age, marital status, health status, sleep quality, relationships with children, frequency of meeting with children, and social support were included in the range of control variables.

The method of instrumental variables (IVs) can be used when standard regression estimates of the relation of interest are biased because of reverse causality, selection bias, measurement error, or the presence of unmeasured confounding effects. The central idea is to use a third, ‘instrumental’ variable to extract variation in the (IV) variable of interest that is unrelated to these problems, and to use this variation to estimate its causal effect on an outcome measure [28]. Instrumental variables need to be highly correlated with endogenous, independent variables, not related to dependent variables, and meet the conditions of being exogenous. In previous studies, some scholars pointed out that Internet use affects loneliness, while others pointed out that loneliness affects the Internet, and there may be a two-way causal relationship between them. This is an endogenous problem that may exist when using MIU to explain individual loneliness. In order to overcome this problem, we used “whether to use mobile phone (smart phone)” and “family monthly post and telecommunications fees (yuan/month)” as instrumental variables. Smart phone ownership is a dichotomous variable, and family monthly post and telecommunications expenses is a continuous variable. We expect a significant positive correlation with mobile Internet use, but no relationship with loneliness.

### 3.2. Empirical Strategy

In this study, we used Stata16 for statistical analysis of the samples. First, we carried out descriptive statistics for relevant variables and conducted a difference analysis between groups based on whether or not to surf the Internet. Second, we used stepwise regression to perform OLS benchmark regression on correlated variables. To address potential endogeneity, we used an instrumental variable approach to explore the effect of MIU on loneliness in the elderly. Then, we used the methods of heterogeneity analysis and quantile regression to explore the specific effects of MIU on loneliness in the elderly in different groups and different quantiles, and tested the robustness of the analysis results. Finally, we explored the pathways through which MIU affects loneliness in the elderly through mediation analysis.

## 4. Results

### 4.1. Descriptive Statistics

Table 1 presents the summary statistics for the variables used in the paper. As we can see, the average loneliness of the elderly is 1.486, which shows that the overall loneliness level of the elderly in China is low, but a considerable number of the elderly still face the problem of high levels of loneliness. From the descriptive statistics of MIU and smartphone ownership, the elderly group using smart phones for mobile Internet only accounts for 11.9% of the total sample. The distribution of variables such as gender, sleep quality, frequency of meeting with children and social support is relatively balanced, but the health status of the elderly is biased due to objective physiological aging. At the same time, contrary to the perception of loneliness, the mean value of the variable of the relationship with their children is high. Further, the mean differences between groups of variables were calculated by whether using mobile Internet, and all reached a significant level. Specifically, the loneliness level of mobile Internet users was significantly lower than that of nonusers (*p* < 0.01). The average age of the respondents in the overall sample is about 68 years old, but the mean difference between groups shows that mobile Internet users are significantly younger than nonusers (65.8 vs. 68.3, *p* < 0.01). At the same time, there are more men than women among mobile Internet users, more with partners than without partners, and their health status, sleep quality, frequency of meeting with their children, and relationships with their children are relatively better. However, mobile Internet users receive less social support than nonusers.

### 4.2. Correlates of Loneliness among Older Adults in China

Based on the correlation analysis of relevant variables, we use OLS regressions to explore the relationship between MIU and older adult loneliness (see Table 2). In OLS model, we use the method of gradually increasing control variables to observe the fitting degree of the model. Model 1 shows that when we do not include any other variables and only regress the two variables of MIU and loneliness, it has a significant negative impact on loneliness (*p* < 0.01), but the model fitting degree is poor. On the basis of model 1, model 2 takes individual demographic characteristics such as gender, age, marriage, health status and sleep quality as control variables into the model, and the goodness of fit of the model has been significantly improved. Model 2 showed that MIU has a significant negative effect on loneliness in the elderly (*p* < 0.01). Considering the influence of family factors, in model 3, in addition to controlling the variables included in model 2, we also added two family factors: relationships with children and frequency of meeting with children. Considering the influence of family factors, in model 3, in addition to controlling the variables included in model 2, we also added two family factors: relationships with children and frequency of meeting with children. The goodness of fit of model 3 was further improved. And MIU has a significant negative impact on the loneliness of the elderly (*p* < 0.01). Finally, the previous literature pointed out that social support is an important influencing factor for loneliness. Therefore, compared with model 3, model 4 adds social support variables as control variables for regression. Model 4 showed that MIU has a significant impact on reducing their loneliness to the level of 1%, and the loneliness of the elderly who do not use the Internet is 8.9% higher than that of the elderly who use the mobile Internet, which shows that the use of mobile Internet can help the elderly alleviate the loneliness. From model 1 to model 4, the fitting results of the model are gradually improving, which proves the rationality of the inclusion of control variables.

As for other control variables, our regression results showed that marital status, health status, sleep quality, frequency of meeting with children, relationships with children and social support were negatively correlated with the elderly’s perception of loneliness. Health status and sleep quality reflect the physiological function of the elderly. A large number of studies have confirmed that health has a significant negative impact on older adult loneliness. The specific mechanism is that the elderly groups with poor health objectively face the problems of limited social communication space, communication energy and cognitive ability, and subjectively face problems such as difficulty in self disclosure, alexithymia, and low self-efficacy, so they have a higher level of loneliness. A spouse, children, and social factors mainly alleviate loneliness in the elderly by providing them corresponding companionship and support.

### 4.3. Impacts of MIU on Loneliness and Its Distribution

In order to solve the potential endogenous issues, we further used the ownership of smart phones and the monthly communication cost of families as the instrumental variables for MIU. Before using the IV approach, we regressed the two instrumental variables with elderly loneliness respectively, so as to explore whether smartphone ownership and family monthly communication expenses have a direct impact on elderly loneliness. Considering the possible multicollinearity between variables, we also calculated the variance inflation factor (VIF). Even if it included MIU and two IVs, none of the indicators had a VIF value greater than 2, indicating that there was no multicollinearity problem between our variables. As shown in the first three columns of Table 3, the results in column 1 show that the family monthly communication fee is not associated with loneliness, and the results in column 2 show that smartphone ownership is not associated with loneliness, and the results in column 3 show that even when both instrumental variables are regressed on loneliness, they are still not associated with loneliness. Therefore, there is no evidence that our two instrumental variables directly affect loneliness in old age, so further analysis can be performed.

The 2SLS estimates are then reported in columns 4 and 5 of Table 3. The first-stage regression results show that both IVs have a positive impact on MIU (column 4). That is, the condition that the instrumental variable is highly correlated with the independent variable is satisfied. The Kleibergen–Paap F statistic has a value of 288.36 (*p* < 0.01), much more than 10, rejecting the null hypothesis of weak instruments. The Hansen J test for instrument exogeneity scored a value of 0.027 (*p* = 0.87 > 0.1), confirming the validity of our chosen instruments. The second-stage showed that after considering the possible endogenous problems in the study and dealing with them, the impact of MIU on elderly loneliness was still significantly negative (*p* < 0.05), and the estimated coefficient of the model increased compared with the estimated coefficient of the benchmark regression results. In contrast, the results of the re-estimation using the instrumental variable method are more accurate and reliable. Specifically, MIU can alleviate the loneliness of the elderly by 0.331 percentage points. The loneliness of the elderly using mobile Internet was 33.1% lower than that of the elderly without mobile Internet. This shows that for the elderly who uses mobile Internet, it can help reduce loneliness by 22.3% (The mean value of loneliness in Table 1 is 1.486; 0.223 = −0.331/1.468). This reflects the empowerment of mobile Internet use for the elderly, which is of great value.

In the benchmark regression model, we mainly investigated the causal relationship between MIU and “average” loneliness, which may mask the large heterogeneity. Therefore, in order to understand the relationship between mobile Internet use and elderly loneliness more comprehensively, we used quantile a regression model to test the impact of mobile Internet use on elderly loneliness at different quantiles. Figure 2 shows the quantile regression coefficient of the impact of MIU on loneliness in the elderly. The quantile interval we set is 0.1–0.95, and the interval in the middle is 0.01. Figure 2 shows that the impact of mobile Internet use on loneliness in the elderly is not linear. When the loneliness of the elderly is below the 0.74 quantile, its value does not change regardless of the use of mobile Internet. In this quantile range, the MIU had no effect on the loneliness of the elderly. When the elderly loneliness is above the 0.74 quantile, the MIU has a significant negative impact on the elderly loneliness in this quantile, and the impact effect gradually increases with the increase of the quantile. This shows that under the control of other relevant variables, the elderly with higher loneliness perception can benefit more from the use of mobile Internet. That is, only when loneliness perception reaches a high score, mobile Internet use has a significant negative impact on it. At the same time, the higher the loneliness of the elderly group, the higher the effectiveness of MIU in alleviating their loneliness.

### 4.4. Heterogeneity Analysis

In order to better understand the conditions under which MIU affects the loneliness of the elderly, we used 2SLS to analyze the heterogeneity. Table 4 presents the heterogeneity analysis results of MIU on loneliness in the elderly. The results show that after controlling the covariates, the impact of mobile Internet use on elderly loneliness varied with age, marital status, and education level. In terms of age, the sample of this study selected the elderly group aged 60–90 and divided it into three age groups for regression. Finally, we found that MIU significantly alleviated the loneliness of the elderly aged 71 to 80 (*p* < 0.05), and the effect on the other two age groups was not significant.

In terms of age, the sample of this study selected the elderly group of 60–90 years old, and divided them into three age groups for regression. Finally, we found that MIU had a significant alleviation effect on the loneliness of the elderly in the age group of 60 to 70 years old, and the effect was statistically significant (*p* < 0.05); no significant effects on the elderly group were seen in the other two age groups. The academic community usually divides the elderly group into three categories: the younger elderly, the middle-aged elderly and the older elderly. Corresponding to this study, the use of mobile Internet has no significant effect on the loneliness of the middle-aged and older elderly people but has a significant alleviating effect on the loneliness of younger-aged elderly people. This may be due to the fact that the younger group of the elderly have just entered old age, their cognitive ability and learning ability have not yet completely declined, their ability to learn to use the mobile Internet is relatively strong, and there are more young elderly people who actually use the mobile Internet. Additionally, they can master more information, communication, entertainment and other related functions, and the mobile Internet has a complementary and beneficial effect on them, which can help them fight against loneliness. The middle-aged and elderly groups may feel a higher sense of loneliness due to the deterioration in physiological functions and the death of their peers. However, it is very difficult for them to use the mobile Internet under the limitation of age and cognitive level, so the number of people who use the mobile Internet objectively is small, and the effect of alleviating loneliness is not obvious. Therefore, it is necessary to pay attention to the difficulties that exist in the use of mobile Internet by the middle-aged and older elderly groups and help them overcome these difficulties so that MIU can help alleviate the loneliness of the middle-aged and older elderly groups.

In terms of marital status, the alleviating effect of mobile Internet use on loneliness was significant only for the elderly with a partner (*p* < 0.05) but not for the elderly without a partner. As the saying goes, “ Young couple, old companion”, and partners play a very important role in individuals’ later lives and play an important role in providing social support and alleviating loneliness. Previous studies have pointed out that older age groups with partners are more likely to master Internet skills due to social support and peer influence than that without partners [29]. Old adults with partners can use mobile Internet more proficiently and healthily, and thus can be positively affected by mobile Internet use. For the elderly group without a partner, it is more likely that they cannot use the mobile Internet very smoothly, or they are overly dependent on mobile Internet use due to the lack of companionship, and excessive use of the Internet may increase loneliness. Therefore, mobile Internet use has a more significant effect on the elderly group with a partner and has no significant effect on the elderly group without a partner.

In terms of education level, mobile Internet use has a significant negative impact on the elderly group with education level of junior high school and below (*p* < 0.1) and has little effect on the elderly group with education level above junior high school. Due to the limitations of social conditions at that time, the education level of the elderly group in the current society is generally low, so we treat the education level as a dichotomous variable of “junior high school and below” and “junior high school and above”. The level of education has a greater impact on the income and social status of individuals. Generally speaking, individuals with higher education levels are in an advantageous position in society, obtain more social resources and social support, and have a relatively low sense of loneliness. At the same time, the older group with a higher education level may have early access to the Internet due to work and other needs, and mobile Internet access is a “natural” change for them. This “subtle” use may have little effect on an individual’s perception of loneliness. However, for the elderly group with a lower education level, they have few social resources themselves, and the level of loneliness perception may be higher. When the mobile Internet enters their lives, it undoubtedly expands the scope of their lives and social interactions and can provide them with corresponding social support and enrich their lives. Therefore, it can further alleviate their original loneliness, and the level of loneliness will be lower than before.

### 4.5. Sensitivity Analysis

The above results all indicate that there is a clear causal association between mobile Internet use and loneliness in the elderly. For the elderly group who use smartphones to surf the Internet, their loneliness is reduced accordingly. In order to further test the robustness of the estimated results, we conducted a battery of sensitivity tests on the estimated results. Table 5 presents the relevant results of the sensitivity tests.

First, we used different models to estimate this research question. Through OLS regression and IV-OProbit model regression results, we found that MIU still had a significant negative impact on loneliness in the elderly. In OLS regression, the influence coefficient of MIU on loneliness in the elderly is −0.089, which is significant at the 0.01 level. In the IV-OProbit model, the influence coefficient of MIU on loneliness in the elderly is −0.5, which is significant at the 0.1 level. Second, we examined the robustness of the results under different sample constraints. We changed the age range of the sample group from the original 60–90 years old to 60–80 years old. The influence coefficient is −0.297, which is significant at the 0.1 level and which is more consistent with the estimated results in the full sample. Finally, we expanded the scope of the independent variables, and combined the two items of “whether to surf the Internet on a mobile phone” and “whether to surf the Internet on a computer” into one variable, that is, “whether to surf the Internet”. The regression results show that whether using Internet has a significant negative impact on loneliness in the elderly, its influence coefficient is −0.316, which is significant at the 0.05 level, and still obtained results similar to the previous regression analysis. These all indicate that the estimation results obtained by the model we used are relatively robust.

At the same time, we used the quantile regression model to test the above four models and analyze the impact of mobile Internet use on loneliness in the elderly at different quantiles under different robustness test models. Figure 3 presents a plot of the quantile regression coefficients of the effects of mobile Internet use on loneliness in the elderly in the four models. We set the quantile interval to be 0.15–0.95, and the interval in between is 0.1. The results are roughly similar to those obtained by using the two-stage least squares method above. When the loneliness of the elderly is below the 0.75 quantile, the effect of mobile Internet use on the loneliness of the elderly is 0. When the elderly loneliness is above the 0.75 quantile, mobile Internet use has a significant negative impact on the elderly loneliness in this quantile, and the effect increases gradually with the increase of the quantile. This also verifies that the model we used and its results are robust.

## 5. Mechanisms

As pointed out in the previous literature analysis, in different situations, Internet use may not only alleviate the level of individual loneliness, but also may increase individual loneliness, so it is of great significance to clarify the path through which mobile Internet use alleviates the loneliness of the elderly. The theory of social–emotional selection emphasizes that as individuals enter old age, they are more willing to manage a network of family members and close friends and constantly shrink their other social networks to expand positive emotions and reduce negative social interactions [30]. This fully reflects the importance of family support and strong ties to the elderly. We selected four variables—parent–child relationship, social interaction (divided into face-to-face interactions and online interactions), and child support—as mediating variables to analyze the mediating effect and to study the mechanism of mobile Internet use affecting loneliness in the elderly and analyze its action path.

Table 6 shows the results of the bootstrap mediation effect test. It can be seen from the table that after controlling for other related variables, the confidence intervals of the three variables of parent–child relationship, face-to-face interaction, and child support do not contain 0. It shows that the mediating effect of these three variables is significant, and they all play a partial mediating effect between mobile Internet use and loneliness in the elderly. Specifically, in the first path, the MIU/parent–child relationship/senior loneliness effect is −0.0968, of which the mediating effect accounts for 7.54% of the total effect; in the second path, the use of mobile Internet for to face-to-face interaction/senior loneliness effect is −0.1204, and the mediating effect accounted for 25.75% of the total effect; and in the third path, the mobile Internet use/child support/senior loneliness effect is −0.0895, of which the mediating effect accounts for 10.39% of the total effect. In the test of the mediating effect of the variable online interaction with children, the confidence interval contains 0, and thus the mediating effect is not significant.

Figure 4 is a map of the effect of mobile Internet use and loneliness in the elderly. It can be seen from the figure that the use of mobile Internet significantly improves the parent–child relationship between the elderly and their children, as well as online and offline social interactions, and these three variables have a significant negative impact on loneliness in the elderly. Therefore, it can be considered that the use of mobile Internet enables the elderly to have more intergenerational communication and interaction with their children, and through the enhancement of intimacy, the loneliness of the elderly is alleviated. In further analysis, we found that the use of mobile Internet has the most obvious effect on improving online communication with children, but the effect of online communication with children on loneliness in old age is far less than that of the parent–child relationship and face-to-face communication; at the same time, there is no mediating effect between mobile Internet use and loneliness. Therefore, although social interaction and social participation can significantly alleviate loneliness in the elderly, they are mostly achieved through actual social interaction. This reflects from the side that the current Chinese elderly are still more accustomed to offline face-to-face interaction. Most of the alleviation of loneliness by using the mobile Internet is achieved by enhancing social support in real life, or supplementing the social support that is lacking in real life. At the same time, we also found that mobile Internet use was significantly negatively correlated with the care support provided by children to parents, while child support was significantly positively correlated with loneliness in old age. This may be due to the fact that children are influenced by the mentality of balanced resource investment, and the use of mobile Internet increases the online communication between adult children and elderly parents. as some tasks can be done directly online, and therefore supporting elderly parents online thus reduces offline care. It is also possible that the elderly who use the mobile Internet are in better health, while the elderly who receive more child-based care have poorer health and are older, so they may feel lonelier.

## 6. Conclusions and Discussion

This study uses representative data to verify the effect of mobile Internet use on loneliness in the elderly, and at the same time, we explored the non-passing effect of mobile Internet use in different parts of the distribution of loneliness in the elderly, further studying the mechanism behind the MIU and loneliness gradients. We believe that looking at the distribution and heterogeneity of loneliness in older age can help us better understand when and to what extent mobile internet use affects loneliness in older age, rather than just focusing on the ‘average’ effect. To our knowledge, this paper is the first to focus on the causal effects of mobile internet use on the distribution of loneliness in the elderly.

We found that although mobile Internet use was associated with a reduction in loneliness among the elderly, mobile Internet use had a more significant alleviating effect on loneliness among the elderly who were between the age of 71 and 80, had a partner, and had an education below the junior high school level. The differences in the influence of mobile Internet use on loneliness among the young elderly, middle-aged, and older elderly may be caused by the loneliness of different age groups and the different abilities in learning and using mobile Internet. The differences in marital status affected by mobile internet use may be due to the fact that older adults with partners are more likely to master the skills of using mobile internet and enjoy the digital dividends than older adults without partners. Finally, the elderly group with an education level below junior high school is more likely to face both the risk of social isolation and social isolation itself. The use of mobile Internet can help them enrich their daily lives, maintain and expand the scope of communication, and effectively alleviate loneliness. For older groups with higher education levels, they have higher social status and stronger social support, and even without the mobile Internet, the social resources they obtain are sufficient to combat loneliness. Therefore, the use of mobile Internet has no significant alleviation effect on the elderly with higher education levels.

We also found that the effects of mobile internet use varied widely across quantiles of the loneliness distribution. Specifically, the alleviating effect of mobile internet use on loneliness was significant only in the higher quantiles of the distribution, and the higher the quantiles, the stronger the effect. This suggests that mobile Internet use provides a new possibility for the elderly with high levels of loneliness, has a complementary effect on their lives, and can alleviate their loneliness. It also has limited impact on the elderly who already live a rich life, have stable relationship networks, and do not seek to relieve their inner loneliness through the mobile Internet.

To gain an in-depth understanding of how mobile internet use affects loneliness in the elderly, we further analyzed the underlying mechanisms behind it. Our mediation analysis shows that mobile internet use alleviates loneliness among older adults by improving parent–child relationships, increasing face-to-face interactions with children, and reducing care support from children, with all three variables having partial mediating effects between them. However, there is no mediating effect between online interaction with children. This result echoes previous studies, which have shown that strong family ties have a greater positive impact on loneliness among the elderly than other social ties [31]. At the same time, compared with online virtual communication spaces, the elderly still prefer offline, actual face-to-face communication [32]. Mobile Internet use needs to be transformed into actual offline communication and support in order to alleviate the loneliness of the elderly. Taken together, these results highlight the emphasis placed on family relationships among older Chinese populations and underscore the critical role of mobile internet use in promoting family support.

Notably, while we demonstrated that mobile internet use had a statistically significant alleviating effect on loneliness in old age, our findings also showed that smartphone ownership had no direct effect on loneliness in old age. In the sample used in this study, most respondents owned a smartphone (78.3%), but only a lower proportion of older adults used a smartphone to surf the Internet (11.9%). This can be understood from the digital divide faced by older age groups in the digital age. Some scholars divide the digital divide into three types: the digital access gap, the digital use gap, and the digital literacy gap [30]. Existing data show that the vast majority of elderly groups in the current society have crossed the digital access gap. However, there are also difficulties in using digital technology and low digital literacy. If the elderly group owns a smartphone but mainly uses traditional communication functions such as text messages and calls, then having a smartphone does not have a significant effect on the sense of loneliness in the elderly. Smartphones can play a role in relieving the loneliness of the elderly only when the elderly group crosses the usage gap and the literacy gap and replaces traditional communication channels with new functions such as social networking services based on smartphones using mobile Internet. On the other hand, most young people turn to the mobile Internet for daily interaction, and the communication gap between the elderly who rely on the traditional functions of mobile phones and other family members, such as young people, has widened. The existence of the digital divide itself also makes older groups who have not mastered digital technology feel social isolation and leads to increased loneliness [33]. In short, due to the different functions in the use of smartphones, the elderly experience different effects on loneliness. The use of mobile Internet can help the elderly to alleviate loneliness but only using traditional communication functions does not. From this perspective, important ways to alleviate the loneliness of the elderly are to help the elderly group adapt to technological development, bridge the digital divide, strengthen intergenerational and interpersonal communication, and maintain or expand the relationship network of the elderly group.

Our findings have policy implications and social value. In current Chinese society, the sense of loneliness among the elderly is relatively high, and the pension status of the whole society shows the characteristics of focusing on material pensions, with less concerned about the emotional needs of the elderly. At the same time, the digital literacy of the elderly group is low, it is difficult to integrate into the digital society, and the overall situation is in a disadvantaged position. However, in recent years, the Chinese government has gradually improved its care for the elderly and attached great importance to the rights and interests of the elderly. In November 2020, the General Office of the State Council issued the “Implementation Plan on Effectively Solving the Difficulties of the Elderly in Using Intelligent Technology”, reflecting the country’s concerns and policy changes regarding the social integration of the elderly in the digital age. This study demonstrates the alleviating effect of mobile Internet use on the loneliness of the elderly. Therefore, it is reasonable and important for the government to take action to help the elderly group integrate into the digital society and alleviate the loneliness of the elderly.

Individual factors affecting loneliness (such as gender, age, education level, etc.) are relatively fixed and difficult to change in the short term, while social factors affecting loneliness experience great changes. With the rapid dissemination and development of information and communication technologies, especially the popularity of smartphones and mobile Internet may have great potential to improve the sense of loneliness in the elderly. We can increase the proportion of the elderly using mobile Internet through intervention and advocacy, so as to alleviate the loneliness of the elderly and improve the mental health of the elderly. In view of the current social situation, this paper puts forward the following suggestions. First, promote the aging-appropriate transformation of intelligent equipment. Considering the particularity of the elderly’s vision, their speed of searching for and understanding information is slower than that of the young, and so on, so the design interface of some smart devices and programs often used by the elderly should be developed to promote a simplified version of the page presentation, such as laying out large fonts, increasing the distance between characters, minimizing the content of each screen page, adding more audio and video materials, etc., thus attracting the interest of the elderly in the login interface and optimizing the effect of human–computer communication. Second, improve the digital literacy of the elderly. In the process of gradually integrating the elderly into the digital society, it is difficult to avoid risks such as Internet addiction, Internet fraud, and dissemination of false information. Moderate use of mobile Internet can alleviate the loneliness of the elderly, but excessive dependence on the Internet may also lead to increased loneliness, and this requires seniors to have high digital literacy to deal with these problems that may arise. Therefore, it is also necessary to focus on improving the digital literacy of the elderly, improving their ability to distinguish various types of information, and reducing the potential risks of digital technology. Third, advocate digital feedback in the family. Digital feedback and family support from family members can help the elderly acquire corresponding Internet skills more quickly and smoothly. At the same time, digital feedback increases the communication between the elderly and other family members, narrows the generation gap within the family, and harmonizes family relationships, which can also reduce the sense of loneliness in the elderly in the process. Finally, children should accompany the elderly more often. Our research found that variables such as frequency of seeing children and relationship with children can significantly reduce loneliness in older adults, and the use of mobile Internet also partially relieves the sense of loneliness in the elderly by increasing the actual communication among family members. Therefore, the actual social interaction between family members is still important to the elderly, and children and younger generations must have the subject consciousness of visiting and accompanying the elderly groups in the family. In addition, since the alleviating effect of mobile Internet use is more significant for the elderly with high levels of loneliness, it is necessary to pay more attention to the elderly with high levels of loneliness when encouraging the elderly to participate in Internet activities.

There are inevitably some design deficiencies and limitations in this study. Given the particularity of our research themes, it is difficult for us to find completely external and objective items (such as objective policy changes and other factors) as instrumental variables for mobile Internet usage. Therefore, the validity of our two instrumental variables (smartphone ownership and monthly household communication costs) may be questioned. For example, our two instrumental variables maybe have a direct effect on loneliness in old age, rather than an indirect effect through mobile internet use. Although studies have shown that smartphones may have a direct impact on loneliness [34,35] due to the generally low participation in digital technology among older groups, we therefore hypothesized that it had no significant direct effect on loneliness in the elderly. We demonstrated the exogeneity and validity of the two instrumental variables selected in this study by directly controlling the two instrumental variables and conducting robustness tests. On the other hand, given that the OLS estimates and the instrumental variables method estimates are relatively close, we believe that the findings are not seriously affected by endogeneity, and the negative impact of mobile Internet use on loneliness in the elderly can be explained as a causal relationship. Second, this study only used a single indicator to measure loneliness, although it is reasonable to some extent, it is not comprehensive enough. Finally, this study only used MIU as an independent variable to explore its effect on loneliness in the elderly. In fact, due to the frequency of Internet use and the type of online activities (study, social, entertainment, and business), it will also have different effects on loneliness in the elderly. Therefore, in future research, the research content can be further refined to explore how the behavioral differences in the use of the Internet affect loneliness.

## Figures and Tables

**Figure 1 ijerph-19-05575-f001:**
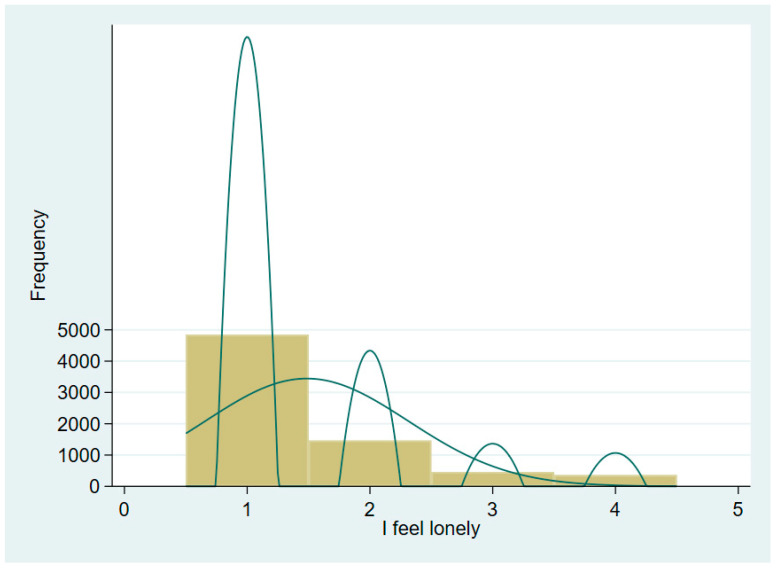
Histogram for loneliness.

**Figure 2 ijerph-19-05575-f002:**
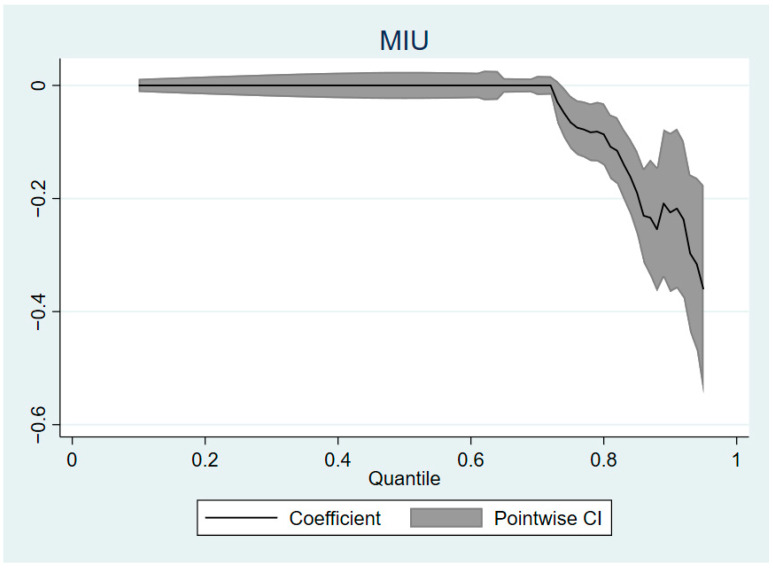
Quantile regression coefficient diagram.

**Figure 3 ijerph-19-05575-f003:**
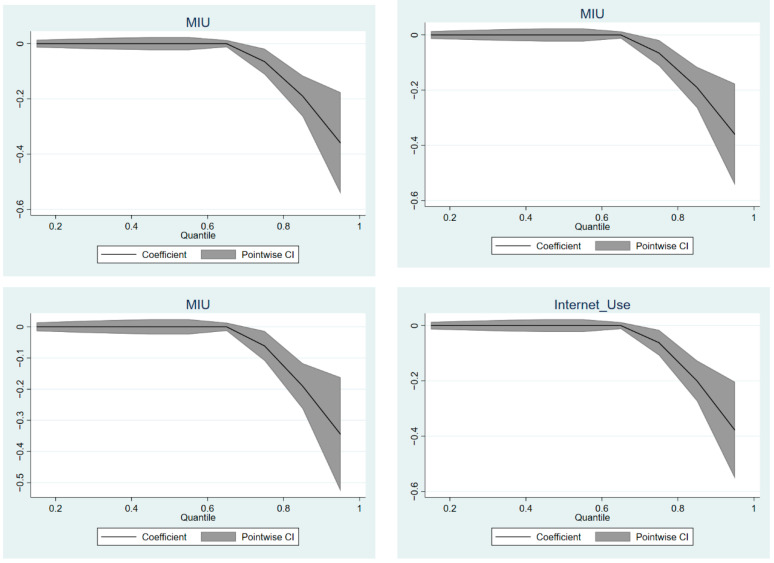
Quantile regression coefficient plots of sensitivity analysis.

**Figure 4 ijerph-19-05575-f004:**
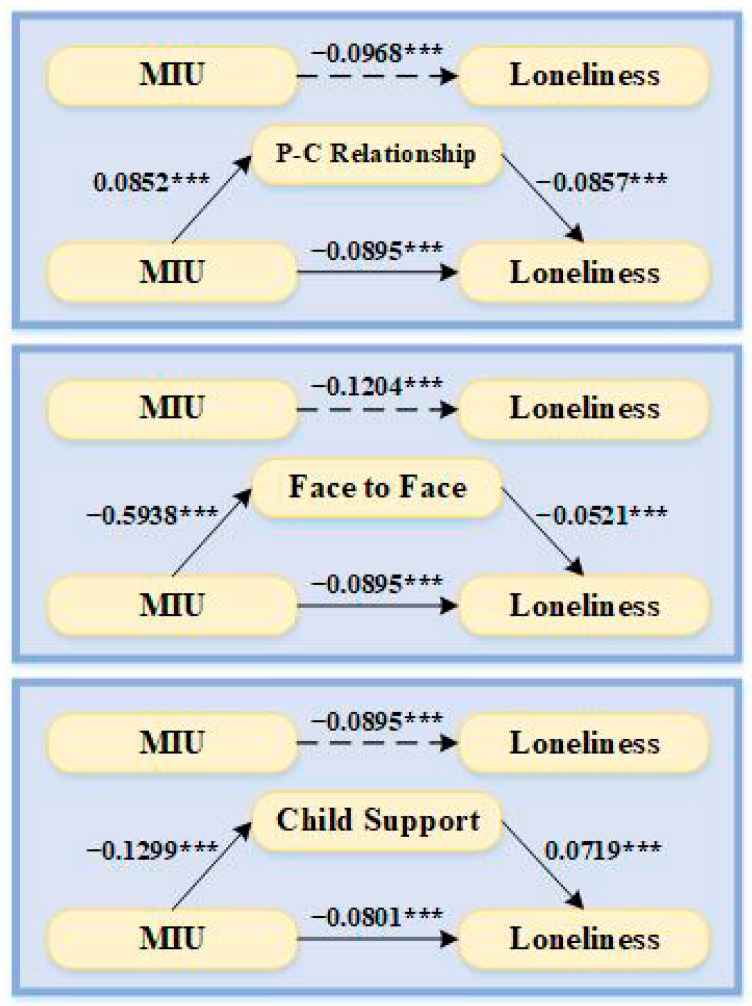
Mediation path map. Note: *** *p* < 0.01.

**Table 1 ijerph-19-05575-t001:** Descriptive statistics.

Variables	N	Full Sample		By MIU Status (Mean)		Diff.
		Mean	S.D	Non-User	User	
Loneliness	7128	1.486	0.826	1.508	1.324	0.184 ***
MIU	7128	0.119	0.323	-	-	-
Gender	7128	0.504	0.500	0.492	0.591	−0.099 ***
Age	7128	68.071	6.179	68.376	65.811	2.565 ***
Married	7128	0.175	0.380	0.184	0.105	0.079 ***
Health	7128	2.499	1.230	2.486	2.599	−0.113 **
Sleep quality	7128	3.049	1.036	3.034	3.160	−0.126 ***
Parent-child relationship	7128	4.311	0.711	4.296	4.421	−0.125 ***
Frequency	7128	4.382	1.779	4.312	4.898	−0.586 ***
Social support	7128	1.759	0.624	1.773	1.655	0.118 ***
Smartphone ownership	7128	0.783	0.412	0.755	0.995	−0.241 ***
Family monthly communication fee	7128	170.122	220.278	160.887	238.704	−77.818 ***

Note: ** *p* < 0.05; *** *p* < 0.01.

**Table 2 ijerph-19-05575-t002:** OLS estimates of the effect of MIU on loneliness among older adults.

	Dependent Variable: Loneliness			
Variables	(1)	(2)	(3)	(4)
MIU	−0.184 ***	−0.128 ***	−0.086 ***	−0.089 ***
	(−7.56)	(−5.45)	(−3.70)	(−3.84)
Gender		0.056 ***	0.036 *	0.036 *
		(2.95)	(1.92)	(1.92)
Age		−0.005 ***	−0.005 ***	−0.005 ***
		(−2.88)	(−2.95)	(−2.74)
Married		0.573 ***	0.569 ***	0.566 ***
		(18.07)	(18.12)	(18.02)
Health		−0.049 ***	−0.042 ***	−0.042 ***
		(−5.96)	(−5.17)	(−5.13)
Sleep quality		−0.181 ***	−0.175 ***	−0.175 ***
		(−16.67)	(−16.34)	(−16.31)
Frequency of meeting children			−0.052 ***	−0.052 ***
			(−10.18)	(−10.14)
Parent–child relationship			−0.087 ***	−0.086 ***
			(−6.28)	(−6.14)
Social support				−0.029 *
				(−1.86)
Constant	1.508 ***	2.372 ***	2.953 ***	2.972 ***
	(141.26)	(20.30)	(22.34)	(22.37)
N	7128	7128	7128	7128
*R* ^2^	0.005	0.138	0.158	0.159

Note: * *p* < 0.1; *** *p* < 0.01; values with * represent regression coefficients, and the values in parentheses below represent the corresponding t-values.

**Table 3 ijerph-19-05575-t003:** Test for the validity of the two instruments and the 2SLS estimates.

	Loneliness	Loneliness	Loneliness	MIU	Loneliness
	OLS	OLS	OLS	2SLS-First Stage	2SLS-Second Stage
Variables	(1)	(2)	(3)	(4)	(5)
Family monthly communication fee	−0.000		−0.000	0.0001 ***	
	(−0.88)		(−0.87)	(5.69)	
Smartphone ownership		−0.030	−0.029	0.1239 ***	
		(−1.28)	(−1.27)	(23.74)	
MIU	−0.087 ***	−0.084 ***	−0.082 ***		−0.334 **
	(−3.07)	(−2.92)	(−2.84)		(−1.99)
Constant	2.980 ***	3.024 ***	3.031 ***	0.1518 ***	3.071 ***
	(24.09)	(23.30)	(23.31)	(3.12)	(20.83)
Controls	Yes	Yes	Yes	Yes	Yes
Hansen J statistic				0.032	
Kleibergen–Paap F statistic				289.868 ***	
N	7128	7128	7128	7128	7128
*R* ^2^	0.159	0.159	0.159	0.068	0.150

Note: ** *p* < 0.05; *** *p* < 0.01; values with * represent regression coefficients, and the values in parentheses below represent the corresponding t-values.

**Table 4 ijerph-19-05575-t004:** Heterogeneity analysis results.

	Age			Married		Education	
	60–70	71–80	81–90	Yes	No	Junior High School and Lower	Above Junior High School
MIU	−0.374 **	−0.164	−1.326	−0.345 **	−0.775	−0.456 *	0.148
	(−2.18)	(−0.46)	(−0.70)	(−2.16)	(−0.92)	(−1.89)	(0.58)
Constant	2.328 ***	2.433 ***	2.220 ***	2.505 ***	4.178 ***	3.138 ***	2.262 ***
	(38.85)	(21.24)	(6.19)	(17.15)	(8.74)	(19.79)	(6.34)
Controls	Yes	Yes	Yes	Yes	Yes	Yes	Yes
N	4965	1821	342	5880	1248	6318	810
*R* ^2^	0.127	0.188	0.047	0.071	0.091	0.143	0.171

Note: * *p* < 0.1; ** *p* < 0.05; *** *p* < 0.01; values with * represent regression coefficients, and the values in parentheses below represent the corresponding t-values.

**Table 5 ijerph-19-05575-t005:** Sensitivity analysis.

	(1)	(2)	(3)	(4)
	OLS	IV-OProbit	Aged 60–80	Internet Use
MIU	−0.089 ***	−0.500 *	−0.297 *	
	(−3.84)	(−1.88)	(−1.80)	
Gender	0.036 *	0.050	0.044 **	0.048 **
	(1.92)	(1.49)	(2.15)	(2.34)
Age	−0.005 ***	−0.010 ***	−0.004 **	−0.006 ***
	(−2.74)	(−3.45)	(−2.05)	(−3.12)
Married	0.566 ***	0.789 ***	0.582 ***	0.562 ***
	(18.02)	(19.16)	(17.54)	(17.78)
Health	−0.042 ***	−0.080 ***	−0.041 ***	−0.042 ***
	(−5.13)	(−6.22)	(−4.99)	(−5.07)
Sleep quality	−0.175 ***	−0.269 ***	−0.171 ***	−0.173 ***
	(−16.31)	(−17.47)	(−15.78)	(−16.17)
Frequency of meeting children	−0.052 ***	−0.085 ***	−0.049 ***	−0.047 ***
	(−10.14)	(−8.08)	(−7.69)	(−7.46)
Parent–child relationship	−0.086 ***	−0.135 ***	−0.086 ***	−0.081 ***
	(−6.14)	(−6.19)	(−5.89)	(−5.60)
Social support	−0.029 *	−0.043 *	−0.041 **	−0.037 **
	(−1.86)	(−1.71)	(−2.45)	(−2.21)
Internet Use				−0.316 **
				(−1.99)
Constant	2.972 ***		2.988 ***	3.062 ***
	(22.37)		(18.67)	(21.03)
N	7128	7128	6786	7128
*R* ^2^	0.159		0.157	0.150

Note: * *p* < 0.1; ** *p* < 0.05; *** *p* < 0.01; values with * represent regression coefficients, and the values in parentheses below represent the corresponding t-values.

**Table 6 ijerph-19-05575-t006:** Bootstrap mediation test.

Effect		Coef.	SE	Bootstrap 95% CI	Proportion
Parent–child relationship	Direct effect	−0.0895	0.0239	−0.1364~−0.0426	7.54%
	Indirect effect	−0.0073	0.0025	−0.0122~−0.0024	
Face-to-face interaction	Direct effect	−0.0895	0.0235	−0.1356~−0.0434	25.71%
	Indirect effect	−0.0309	0.0049	−0.0405~−0.0214	
Online communication	Direct effect	−0.0802	0.0240	−0.1272~−0.0332	10.37%
	Indirect effect	−0.0093	0.0049	−0.0189~0.0003	
Child support	Direct effect	−0.0801	0.0230	−0.1253~−0.0350	10.43%
	Indirect effect	−0.0093	0.0028	−0.0148~−0.0039	

## Data Availability

This paper uses data from the Chinese Family Panel Studies (CFPS), a nationally representative longitudinal survey funded by Peking University and the National Natural Science Foundation of China and initiated by the Institute of Social Sciences of Peking University.

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
