# Peer review of "Does Mobile Internet Use Affect the Loneliness of Older Chinese Adults? An Instrumental Variable Quantile Analysis"

_ijerph, 2022, doi:10.3390/ijerph19095575_

Round 1

Reviewer 1 Report

The paper submitted for review Does Mobile Internet Use Affect the Loneliness of Older Chinese Adults? An Instrumental Variable Quantile Analysis discusses a very important issue of the impact of mobile Internet on the feeling of loneliness of the elderly in China. The research results obtained by the authors indicate that loneliness was 33% lower in people using the Internet than in those who did not use the Internet. Mediation analysis also demonstrated that the use of mobile Internet improves the offline relationships between seniors and their children. The methods and tools used in the research do not arouse methodological objections. However, the English language needs to be corrected, in particular typing errors, such as "Absteact" (p. 1), "Gemder" (Table 4) and punctuation "(...) and lead to an increase in individual loneliness. (Kraut et al., 1998; Nie et al., 2001)." (p. 4). In view of the above, I recommend the work for publication after correction of the language.

Author Response

Thank you very much for your valuable comments.

  1. The English language needs to be corrected, in particular typing errors, such as "Absteact" (p. 1), "Gemder" (Table 4) and punctuation "(...) and lead to an increase in individual loneliness. (Kraut et al., 1998; Nie et al., 2001)." (p. 4). In view of the above, I recommend the work for publication after correction of the language.

Reply: We are very sorry for the spelling mistakes in the article. We have corrected the language errors in the article.

Reviewer 2 Report

Dear Editor,

Thank you for the opportunity to review this manuscript. Herewith my feedback:

  1. The authors integrate information that typically should be in the Discussion section in their Method and Data Analyses sections. For example, where they attempt to explain in the data and variable sections why the mean loneliness score is lower than expected.
  2. There are numerous examples where the authors assume that the readers would understand their reasoning. Here are a few examples:
    • What were the authors expecting in terms of the mean loneliness score if it is lower than what they anticipated?
    • What is meant with instrumental variables?
    • What is meant by possible endogenous problems?
  3. A major cause for concern is the heavy reliance on a single item measure of resilience. In this regard, it has long been demonstrated that single item measures cannot be used for the assessment of complex constructs. When used as a single item measure, it amounts to self-diagnosis of a psychological problem. Self-diagnosis of psychological problems is problematic as respondents are not qualified to make such considerations and may over or understate their symptoms. Generally, in terms existing literature, there are three major concerns with using single items scales: they do not adequately capture the construct (i.e., there is low construct validity); have fewer points of discrimination (i.e., sensitivity) and lack a measure of internal consistency reliability.
  4. Related to the above is concerns about the operationalization of loneliness and the subtle difference between the frequency of loneliness and the subjective experience of loneliness. Based on the ordinal scale, which includes “less than one a day”, “three or four days”, this variable appears to reflect the frequency of occurrence (i.e., how often one feels lonely).
  5. The explanation of the statistical analysis called “empirical strategy” is unnecessarily technical. It is more useful to know what analytical programme has been used and what analytical technique was implemented. I do not believe that the authors could have analysed a data set of 7128 participants by hand and thus it is better to provide a non-technical explanation of the objective of the analyses.
  6. Another significant concern is the over-reliance on statistical significance testing. Given the large sample size, even small differences will be significant. It is therefore more useful to provide effect size estimates rather than just p values. Related to this is the number of comparisons which raises the multiple comparison problem where the probability of finding a significant result that is due to chance increases. The authors are encouraged to look at tools such as the false discovery rate calculators that offers a potential correction to the issue of multiple comparisons.
  7. The same issue of reliance on statistical significance applies to correlation. It does not make sense that a co-efficient of .04 can be regarded as meaningfully significant even if it is statistically significant. A coefficient of .04 indicates that the two variables share 0.001% of variance. Most modern software provides confidence intervals and effect sizes for these correlation coefficients.
  8. The Tables generally are not well structured and do not make sense.
    • For example, with regard to the table of intercorrelations, the above and below the diagonal should be mirror images of each other yet in Table 3 the above and below the diagonal differs in several ways – the relationship between loneliness and MIU above the diagonal is -0.06 but below the diagonal the same relationship is -0.07. Also it is convention to report only below the diagonal because they are meant to be mirror images of each other.
    • Table 4: is similarly not well constructed or explained in a table note. I presume that values in the first row are the regression coefficients but I do not know what the values in brackets below the first row refer to. Also report the confidence interval for these regression coefficients.
  9. The authors should explain why the two SLS analyses reported in Table 5 confirms the validity of their chosen instruments. Without sufficient explanation, this is a huge argumentative leap.
  10. What provincial level is being referred to in the note for Table 5?
  11. The section on Mechanisms and in particular the testing of mediation is laudable as authors went beyond the mere cause and effect relationship. However, I would suggest that in figure 4, the authors should add to the diagram the path from MIU to loneliness prior to the inclusion of the mediator and this path can be shown as a dotted line.
  12. In sum, four figures and eight tables raise some concerns in terms of over analyses of data. In light of the concerns around reliance on statistical significance testing as well as the fact that the dependent variable is represented by a single item, there are concerns regarding whether such analyses are justified.

Author Response

Thank you very much for your valuable comments.

  1. The authors integrate information that typically should be in the Discussion section in their Method and Data Analyses sections. For example, where they attempt to explain in the data and variable sections why the mean loneliness score is lower than expected.

Reply: We deleted the information that typically should be in the Discussion section in their Method and Data Analyses sections.

  1. There are numerous examples where the authors assume that the readers would understand their reasoning. Here are a few examples:
    • What were the authors expecting in terms of the mean loneliness score if it is lower than what they anticipated?
    • Reply: In this part, we supplement the literatureson measuring loneliness with the same scale.The mean loneliness score obtained in this study(1.486) is lower than that obtained in the relevant literatures(1.6&1.77). Therefore, we explain this in the article.
    • What is meant with instrumental variables?
    • Reply:In the data and methods section, we added a supplementary description of instrumental variables. The supplementary contents are as follows:

“The method of instrumental variables (IVs) can be used when standard regression estimates of the relation of interest are biased because of reverse causality, selection bias, measurement error, or the presence of unmeasured confounding effects. The central idea is to use a third, ‘instrumental’ variable to extract variation in the (IV) variable of interest that is unrelated to these problems, and to use this variation to estimate its causal effect on an outcome measure(Stock,2001). Instrumental variables need to be highly correlated with endogenous independent variables, not related to dependent variables, and meet the conditions of exogenous.”

  • What is meant by possible endogenous problems?
  • Reply:The possible endogenous problem is that there may be a two-way causal relationship between MIU and loneliness in the elderly. That is, it may be that the use of mobile Internet affects the loneliness of the elderly, it may also be that the loneliness of the elderly affects the use of mobile Internet, or it may affect each other.In this context, the instrumental variable method needs to be used to solve the potential endogenous problems.          We added a detailed explanation in the paper to facilitate readers' understanding.

  1. A major cause for concern is the heavy reliance on a single item measure of resilience. In this regard, it has long been demonstrated that single item measures cannot be used for the assessment of complex constructs. When used as a single item measure, it amounts to self-diagnosis of a psychological problem. Self-diagnosis of psychological problems is problematic as respondents are not qualified to make such considerations and may over or understate their symptoms. Generally, in terms existing literature, there are three major concerns with using single items scales: they do not adequately capture the construct (i.e., there is low construct validity); have fewer points of discrimination (i.e., sensitivity) and lack a measure of internal consistency reliability.

Reply: Our study uses one question in CFPS to evaluate loneliness, which can reflect individual loneliness to a certain extent. At the same time, many studies have adopted one question to assess the loneliness of elderly, and obtained reliable results, so it is reasonable to use a question to assess individual loneliness. Literature that supports our assertion of causation includes:

[1] Song S ,  Song S ,  Zhao Y C , et al. INVESTIGATING THE RELATIONSHIP BETWEEN INTERNET USE AND PERCEIVED LONELINESS AMONG OLDER CHINESE[J]. Innovation in Aging, 2019(Supplement_1):Supplement_1.

[2] Zhou Z ,  Mao F ,  Zhang W , et al. The Association Between Loneliness and Cognitive Impairment among Older Men and Women in China: A Nationwide Longitudinal Study[J]. International Journal of Environmental Research and Public Health, 1943, 16(16).

However, the lack of professional scales such as UCLA to measure loneliness in CFPS is indeed a limitation of this study, so we add this question to the limitations section at the end of the article.

  1. Related to the above is concerns about the operationalization of loneliness and the subtle difference between the frequency of loneliness and the subjective experience of loneliness. Based on the ordinal scale, which includes “less than one a day”, “three or four days”, this variable appears to reflect the frequency of occurrence (i.e., how often one feels lonely).

Reply: Our study uses one question in CFPS to evaluate loneliness, which can reflect individual loneliness to a certain extent. At the same time, many studies have adopted one question to assess the loneliness of elderly, and obtained reliable results, so it is reasonable to use a question to assess individual loneliness. Literature that supports our assertion of causation includes:

[1] Song S ,  Song S ,  Zhao Y C , et al. INVESTIGATING THE RELATIONSHIP BETWEEN INTERNET USE AND PERCEIVED LONELINESS AMONG OLDER CHINESE[J]. Innovation in Aging, 2019(Supplement_1):Supplement_1.

[2] Zhou Z ,  Mao F ,  Zhang W , et al. The Association Between Loneliness and Cognitive Impairment among Older Men and Women in China: A Nationwide Longitudinal Study[J]. International Journal of Environmental Research and Public Health, 1943, 16(16).

However, the lack of professional scales such as UCLA to measure loneliness in CFPS is indeed a limitation of this study, so we add this question to the limitations section at the end of the article.

  1. The explanation of the statistical analysis called “empirical strategy” is unnecessarily technical. It is more useful to know what analytical programme has been used and what analytical technique was implemented. I do not believe that the authors could have analysed a data set of 7128 participants by hand and thus it is better to provide a non-technical explanation of the objective of the analyses.

Reply: We deleted the explanation of statistical analysis in "empirical strategy" and added analytical programme and analytical technique.

  1. Another significant concern is the over-reliance on statistical significance testing. Given the large sample size, even small differences will be significant. It is therefore more useful to provide effect size estimatesrather than just p values. Related to this is the number of comparisons which raises the multiple comparison problem where the probability of finding a significant result that is due to chance increases. The authors are encouraged to look at tools such as the false discovery rate calculators that offers a potential correction to the issue of multiple comparisons.

Reply: In the process of data analysis, we not only rely on the p value, but also judge the analysis results according to the confidence interval and other indicators. The results show that our analysis is reasonable. However, due to the structure of the tables in the paper, some indicators cannot be presented in them. If necessary, we can provide the original data and code we use for analysis in this study.

  1. The same issue of reliance on statistical significance applies to correlation. It does not make sense that a co-efficient of .04 can be regarded as meaningfully significant even if it is statistically significant. A coefficient of .04 indicates that the two variables share 0.001% of variance. Most modern software provides confidence intervals and effect sizes for these correlation coefficients.

Reply: During the research process, the scope of application of correlation analysis was not grasped, and inapplicable errors occurred. After learning, it was found that correlation analysis could not be used for the analysis of categorical variables. Therefore, the correlation analysis is deleted in the article, and the relationship between variables is directly explored through subsequent regression analysis.

  1. The Tables generally are not well structured and do not make sense.
    • For example, with regard to the table of intercorrelations, the above and below the diagonal should be mirror images of each other yet in Table 3 the above and below the diagonal differs in several ways – the relationship between loneliness and MIU above the diagonal is -0.06 but below the diagonal the same relationship is -0.07. Also it is convention to report only below the diagonal because they are meant to be mirror images of each other.
    • Reply:In the subsequent benchmark regression analysis, the relationship between variables is analyzed in detail, so Table 3 and correlation analysis are deleted.
    • Table 4: is similarly not well constructed or explained in a table note. I presume that values in the first row are the regression coefficients but I do not know what the values in brackets below the first row refer to. Also report the confidence interval for these regression coefficients.
    • Reply:We revised the notes of all tables to explain the meaning of each value. The values in the first row are the regression coefficients, and the values in brackets below the first row refer to t-vaule.However, due to the table structure, we cannot add confidence intervals to the corresponding tables. Therefore, the relevant confidence interval tables are attached here.

  1. The authors should explain why the two SLS analyses reported in Table 5 confirms the validity of their chosen instruments. Without sufficient explanation, this is a huge argumentative leap.

Reply: In the explanation part of table 5, the explanation and elaboration on the effectiveness of instrumental variables are added to explain that the two instrumental variables meet the corresponding conditions, which proves that the instrumental variables we selected are appropriate.

  1. What provincial level is being referred to in the note for Table 5?

Reply: "The robust standard errors clustered at the provincial level are shown in parents." This sentence was added by mistake, so we delete it from the article.

  1. The section on Mechanisms and in particular the testing of mediation is laudable as authors went beyond the mere cause and effect relationship. However, I would suggest that in figure 4, the authors should add to the diagram the path from MIU to loneliness prior to the inclusion of the mediator and this path can be shown as a dotted line.

Reply: We added to the diagram the path from MIU to loneliness prior to the inclusion of the mediator and the path is shown as a dotted line. However, due to the different control variables in the analysis of the three intermediary variables, the total effect is different. We add the total effect path to the front of the three intermediary path respectively.

  1. In sum, four figures and eight tables raise some concerns in terms of over analyses of data. In light of the concerns around reliance on statistical significance testing as well as the fact that the dependent variable is represented by a single item, there are concerns regarding whether such analyses are justified.

Reply: We changed the presentation of the figures and tables,we deleted the correlation analysis part in the article, and combined the difference between groups and descriptive analysis into a table for easy understanding. At the same time, the controversial parts of data analysis are changed, and the corresponding explanation and demonstration are provided for the measurement of dependent variables using a single item. After modification, our study is justified in data analysis.This paper discusses the impact of MIU on loneliness and its mechanism, which has theoretical significance and practical concern.

Reviewer 3 Report

Thanks for giving me the opportunities to review this manuscript titled “Does Mobile Internet Use Affect the Loneliness of Older Chinese Adults? An Instrumental Variable Quantile Analysis.” This study analyzed the association between loneliness and mobile internet use based on the data of CFPS 2018 wave. The topic is very important to be explored. However, I also have several suggestions for this manuscript.

  1. As the data analyzed in this study was from the CFPS 2018, we can see it as a cross-sectional design. Because of this, any causal relationship between MIU and loneliness cannot be inferred in this study. The authors should revise this kind of explanations throughout this manuscript.
  2. Loneliness was evaluated by one question. Actually, we have several scales to evaluate loneliness. I think this should be a limitation for this study.
  3. The authors used loneliness as the ID variable. I think using MIU as the ID variable may be easier for us to follow.
  4. I think the data analyses should be revised. As you can see, gender and married status are categorical variable, and it is not right to do the correlation analyses.
  5. There are many mistakes in this manuscript. Please check the manuscript to make sure the word correction. For example, Gemder in Table 4.
  6. The author did lots of analyses, and it make the manuscript a little hard to be followed. I suggested the authors combine or delete some results. Some of them are not necessary.

Author Response

Thank you very much for your valuable comments.

  1. As the data analyzed in this study was from the CFPS 2018, we can see it as a cross-sectional design. Because of this, any causal relationship between MIU and loneliness cannot be inferred in this study. The authors should revise this kind of explanations throughout this manuscript.

Reply: Our study uses the data from CFPS2018, which can indeed be regarded as a cross-sectional design, and the regression analysis with this data can indeed only obtain a correlation rather than a causal relationship. However, the instrumental variable method we use in this article can help overcome potential endogeneity problems such as bidirectional causality, and the causal relationship between two variables can be obtained by using the instrumental variable method.

Literature that supports our assertion of causation includes:

Angrist, J. D., Imbens, G. W., & Rubin, D. B. (1996). Identifcation of causal efects using instrumental variables. Journal of the American Statistical Association, 91(434), 444–455.

Lu H ,  Kandilov I T . Does Mobile Internet Use Affect the Subjective Well-being of Older Chinese Adults? An Instrumental Variable Quantile Analysis[J]. Journal of Happiness Studies, 2021(5).

  1. Loneliness was evaluated by one question. Actually, we have several scales to evaluate loneliness. I think this should be a limitation for this study.

Reply: Our study uses one question in CFPS to evaluate loneliness, which can reflect individual loneliness to a certain extent. At the same time, many studies have adopted one question to assess the loneliness of elderly, and obtained reliable results, so it is reasonable to use a question to assess individual loneliness. Literature that supports our assertion of causation includes:

[1] Song S ,  Song S ,  Zhao Y C , et al. INVESTIGATING THE RELATIONSHIP BETWEEN INTERNET USE AND PERCEIVED LONELINESS AMONG OLDER CHINESE[J]. Innovation in Aging, 2019(Supplement_1):Supplement_1.

[2] Zhou Z ,  Mao F ,  Zhang W , et al. The Association Between Loneliness and Cognitive Impairment among Older Men and Women in China: A Nationwide Longitudinal Study[J]. International Journal of Environmental Research and Public Health, 1943, 16(16).

However, the lack of professional scales such as UCLA to measure loneliness in CFPS is indeed a limitation of this study, so we add this question to the limitations section at the end of the article.

  1. The authors used loneliness as the ID variable. I think using MIU as the ID variable may be easier for us to follow.

Reply: This study attempts to explore the effect of MIU on loneliness in the elderly, so MIU is used as an independent variable and loneliness is used as a dependent variable. Our conclusion shows that MIU has a relieving effect on loneliness in the elderly, which provides a way for us to improve the mental health of the elderly and has positive significance for the current aging society.

  1. I think the data analyses should be revised. As you can see, gender and married status are categorical variable, and it is not right to do the correlation analyses.

Reply: During the research process, the scope of application of correlation analysis was not grasped, and inapplicable errors occurred. After learning, it was found that correlation analysis could not be used for the analysis of categorical variables. Therefore, the correlation analysis is deleted in the article, and the relationship between variables is directly explored through subsequent regression analysis.

  1. There are many mistakes in this manuscript. Please check the manuscript to make sure the word correction. For example, Gemder in Table 4.

Reply: We are very sorry for the spelling mistakes in the article, we have re-checked the article and made corrections

  1. The author did lots of analyses, and it make the manuscript a little hard to be followed. I suggested the authors combine or delete some results. Some of them are not necessary.

Reply: We deleted the correlation analysis part in the article, and combined the difference between groups and descriptive analysis into a table for easy understanding.

Round 2

Reviewer 2 Report

Dear Authors, 

Thank you for engaging with the feedback in a meaningful way. The paper is more refined and I accept it in its present form. 

Sincerely. 

Reviewer 3 Report

Thanks for the responses from the authors. I suggest the authors should be cautious to get the causal conclusion based on the method of instrumental variable quantile analysis.